# Room-temperature high-precision printing of flexible wireless electronics based on MXene inks

Yuzhou Shao [1], Lusong Wei[2], Xinyue Wu[1], Chengmei Jiang[1], Yao Yao [1], Bo Peng[1], Han Chen[1], Jiangtao Huangfu[2], Yibin Ying [1✉], Chuanfang John Zhang [3,4✉] & Jianfeng Ping [1✉]

Wireless technologies-supported printed flexible electronics are crucial for the Internet of Things (IoTs), human-machine interaction, wearable and biomedical applications. However, the challenges to existing printing approaches remain, such as low printing precision, difficulty in conformal printing, complex ink formulations and processes. Here we present a room-temperature direct printing strategy for flexible wireless electronics, where distinct high-performance functional modules (e.g., antennas, micro-supercapacitors, and sensors) can be fabricated with high resolution and further integrated on various flat/curved substrates. The additive-free titanium carbide ($Ti_3C_2T_x$) MXene aqueous inks are regulated with large single-layer ratio (>90%) and narrow flake size distribution, offering metallic conductivity (~6, 900 S cm$^{-1}$) in the ultrafine-printed tracks (3 μm line gap and 0.43% spatial uniformity) without annealing. In particular, we build an all-MXene-printed integrated system capable of wireless communication, energy harvesting, and smart sensing. This work opens a door for high-precision additive manufacturing of printed wireless electronics at room temperature.

---

[1] Laboratory of Agricultural Information Intelligent, School of Biosystems Engineering and Food Science, Zhejiang University, 310058 Hangzhou, China.
[2] Laboratory of Applied Research on Electromagnetics, Zhejiang University, 310027 Hangzhou, China. [3] College of Materials Science & Engineering, Sichuan University, 610065 Chengdu, Sichuan, China. [4] Swiss Federal Laboratories for Materials Science and Technology (Empa), ETH Domain, Überlandstrasse 129, CH-8600 Dübendorf, Switzerland. ✉email: ybying@zju.edu.cn; chuanfang.zhang@empa.ch; jfping@zju.edu.cn

Advances in printed electronics continuously stimulate the scalable and sustainable fabrication of wearable and flexible devices[1–3]. Unlike traditional subtractive processes, direct ink printing offers a viable alternative for rapid, large-scale manufacturing due to its relatively facile, cost-effective procedures, and desirable material compatibility and utilization[4,5]. Nevertheless, with regard to the room-temperature fabrication of flexible electronics, the existing printing approaches are yet far from ideal. The major hurdle comes from ink formulations and printing processes. Most printable inks (metal or carbon-based) either suffer from complex ink formulations (requiring surfactants/rheological modifiers/binders), unsatisfactory intrinsic physical properties (i.e., poor electrical conductivity), or demand lengthy post-treatments (i.e., high-temperature annealing to remove additives)[6,7]. These issues complicate the device's manufacturing process, exclude the low-cost polymeric substrate choices while compromise device printing precision and thereafter properties. On the other hand, the increasing structural complexity of flexible electronics (especially various wireless multi-functional systems) puts higher requirements for direct ink printing technologies, particularly high-precision conformal printing and multi-module integrated manufacturing to avoid time-consuming cumbersome transfer and assembly processes[8,9].

One promising approach is to combine additive-free aqueous conductive inks with extrusion printing technology. Compared with other printing methods, extrusion printing allows high-throughput additive manufacturing without additional masks and accessories, offering greater opportunities in material/substrate choice and printing extensibility (from co-planar to three-dimensional)[10,11]. Nonetheless, while additive-free aqueous conductive inks have been proven promising in simplifying ink formulation and eliminating post-processing, it remains a challenge to endow functional inks with appropriate rheological and electrical properties to achieve room-temperature fabrication of flexible wireless electronics[12,13]. In this regard, as an emerging family of 2D transition metal carbides and nitrides, MXenes, which possess unique properties desirable for functional inks (i.e., metallic conductivity, hydrophilicity, and negative surface charges), offer new possibilities[14,15]. Especially, $Ti_3C_2T_x$ ($T_x$ denotes surface terminations), as the most widely studied MXene, allows controllable formation of stable additive-free aqueous colloidal dispersions without any additives[16,17] and thus has been applied in different devices, such as batteries, micro-supercapacitors (MSCs), triboelectric nanogenerators (TENGs), transistors, sensors, etc.[18–21] However, when it comes to fabricating flexible wireless electronics, little success has been achieved on room-temperature, fine printing precision of component lines with ultrahigh electrical conductivity based on MXene inks. Moreover, feasible protocol of multi-module integrated printing for all-printed wireless devices has been rarely reported so far.

In this Article, we report on the realization of direct printing of flexible wireless electronics at room temperature. Additive-free MXene aqueous inks possess desirable rheological and electrical properties, stemming from large single-layer ratio, high ink concentration, and narrow flake size distribution, to achieve metallic conductivity in high-precision extrusion printing, thereby enabling the efficient fabrication of monolithic flexible systems for energy harvesting, wireless communication, and sensing. The complete demonstration of all-MXene functional electronics powerfully reveals the enormous potential of room-temperature direct MXene printing for large-scale integrated manufacturing of next-generation wearable and flexible wireless electronics.

**Extrusion printable MXene aqueous ink**. We start to describe the MXene ink formulation and characterizations. Figure 1a schematically presents the room-temperature printing strategy for flexible wireless electronics. Additive-free MXene aqueous inks are prepared following a modified minimally intensive layer delamination route, using optimized centrifugation and ultrasonic methods to improve rheological and electrical properties (Supplementary Fig. 1). The as-formulated inks, with a high concentration (~60 mg mL$^{-1}$), contain ultrathin, predominantly

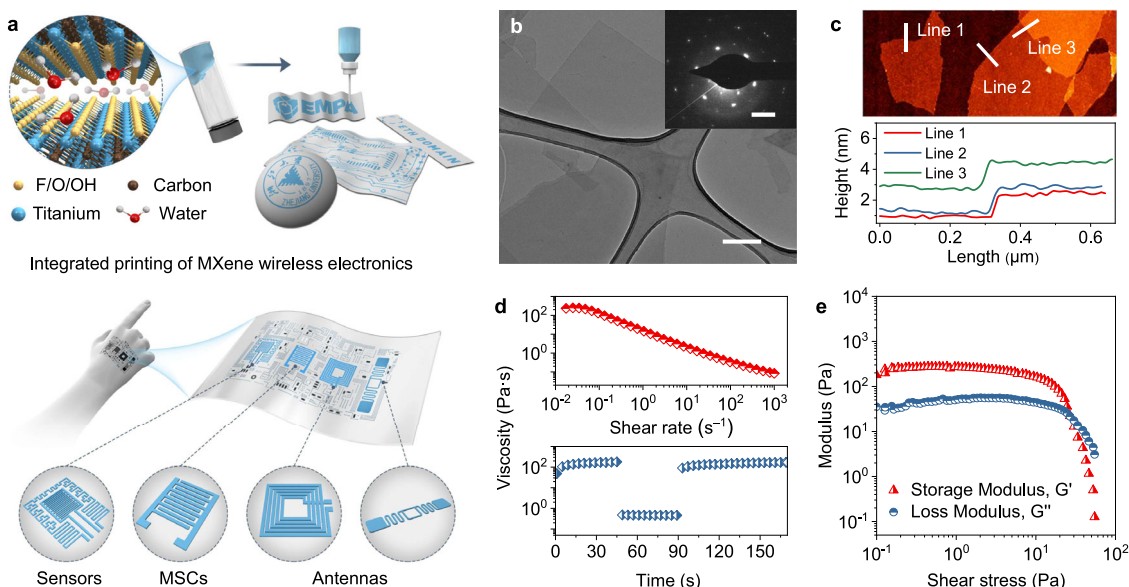

**Fig. 1 Characterizations of printable additive-free MXene aqueous inks. a** Schematic illustration of room-temperature direct printing of additive-free MXene aqueous inks supported on various substrates for flexible wireless electronics, such as sensors, MSCs, antennas, etc. **b** Transmission electron microscopy (TEM) image of $Ti_3C_2T_x$ nanosheets. Scale bar: 200 nm. Inset: the selected area electron diffraction (SAED) pattern. Scale bar: 5 1/nm. **c** Atomic force microscopy (AFM) image of MXene inks and corresponding height profiles. **d** Rheological properties of MXene aqueous inks with viscosity plotted as a function of shear rate (top) and interval shearing time (bottom, alternating the shear rate between 0.1 s$^{-1}$ and 100 s$^{-1}$ to simulate the extrusion process). **e** Storage modulus ($G'$) and loss modulus ($G''$) of MXene aqueous inks versus shear stress.

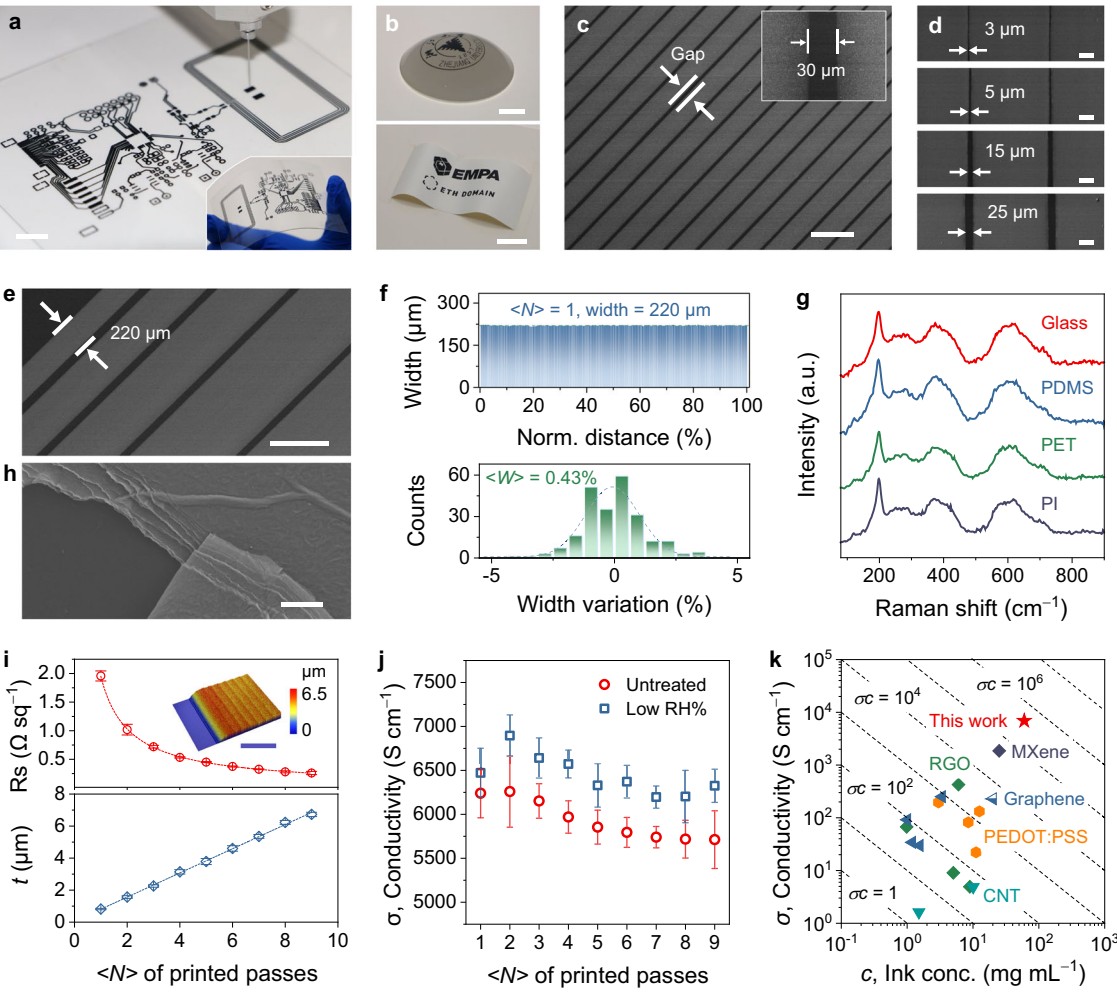

**Fig. 2 Direct printing of viscous MXene aqueous inks. a** Optical image of high-resolution integrated circuits fabricated through direct MXene printing. Scale bar, 10 mm. Inset: the bended MXene circuit. **b** MXene printed "ZJU" and "EMPA" logos on curved surfaces. Scale bar, 20 mm. **c** SEM image of printed periodically MXene lines with a gap of 30 μm. Scale bar, 500 μm. Inset: line gap of 30 μm. **d** SEM images of MXene lines with different gaps from 3 μm to 25 μm. Scale bar, 50 μm. **e** SEM image of the MXene lines of different widths. Scale bar, 500 μm. **f** Corresponding width distribution (top) and variation (<W>, bottom) in **e**. **g** Raman spectrum of MXene films on different substrates. **h** SEM image of the MXene film. Scale bar, 25 μm. **i** Sheet resistance (in red) and thickness (in blue) of MXene films as a function number of printing pass, <N>. Inset: the surface profile of MXene films (<N> = 6). Scale bar, 1 mm. **j** The electrical conductivity changes of MXene films as a function of <N>. The red and blue dots represent that the MXene films were dried under ambient conditions and low humidity, respectively. **k** The comparison of ink conductivity ($\sigma$) and concentration (c) of the MXene ink with other reported printable ink systems.

single-layer $Ti_3C_2T_x$ flakes with the hexagonal atomic arrangement (Fig. 1b and inset), agreeing well with previous reports[21,22]. Those single-layer $Ti_3C_2T_x$ nanosheets possess an average flake size of ~1.6 μm and a thickness of ~1.5 nm (Fig. 1c and Supplementary Fig. 2). Due to high ink concentration, large single-layer ratio (>90%), and narrow flake size distribution, the as-formulated inks showcase desirable shear-thinning viscoelastic properties (viscosity of ~$2.5 \times 10^2$ Pa·s) allowing continuous extrusion and quick solidification (Fig. 1d, e)[23]. Supplementary Figure 3 provides more details regarding the ink rheological characteristics. Notably, these aqueous inks are stable without sedimentation when stored in Ar-sealed bottles in the dark and low temperature (<4 °C) for at least two years, ensuring a sufficient time window for potential ink printing. After removing dissolved oxygen, these aqueous inks are also stable for long periods of time under ambient conditions (Supplementary Fig. 4). Besides, the ink wettability on the substrates are enhanced through plasma treatments to form continuous films and improve the substrate adhesion (See more details in Supplementary Figs. 5–7)[24,25].

**Direct printing of MXene ink.** Room-temperature direct printing was performed by a programmable three-axis pneumatic extrusion dispenser (Fig. 2a and Supplementary Fig. 8). Through digital predefinition with specific line gap and width, diverse patterns/circuits can be efficiently printed on planar or curved surfaces with irregular geometries (such as leaves and fruits, Fig. 2b, Supplementary Fig. 9, and Movie 1). For instance, uniform MXene lines with precise line gaps ranging from 3 to 30 μm can be directly printed (Fig. 2c, d and Supplementary Fig. 10), demonstrating the versatility of the direct printing of fine conductive tracks/circuits. It's worth mentioning that a line gap of 3 μm has yet been reported for direct printing strategies (Supplementary Table 1), revealing the huge potential in fabricating high-resolution, high packing density electronics[8]. In addition, lines with different widths can be printed with an ultrahigh spatial uniformity of 0.43% (Fig. 2e, f), as a result of proper solvent evaporation kinetics and substrate wettability. Measured by an optical profilometer (Supplementary Fig. 11), these MXene lines exhibit elliptical cross-sectional sharp shapes, as benefited from the appropriate ink rheological properties with high G', allowing

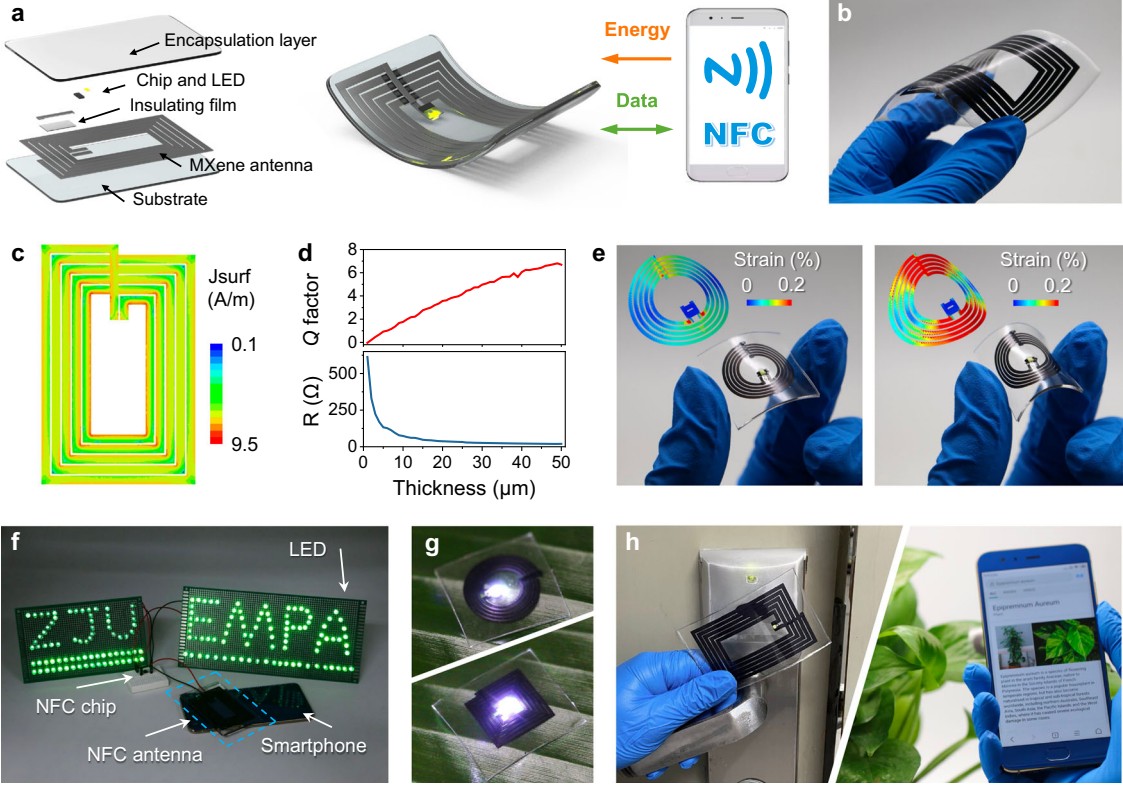

**Fig. 3 Demonstration of MXene NFC devices. a** Fabrication and mechanism of printed MXene NFC tags which communicate wirelessly with the smartphone and capture its power to light up the LED. **b** Photograph of the flexible NFC tag fabricated on PDMS. **c** Surface current distribution (Jsurf) of the NFC antenna at 13.56 MHz. **d** Simulation of the NFC antenna's resistance and Q factor for different thicknesses. **e** Optical images of the curved small-sized NFC tags and corresponding strain distributions (inset) of the flexible antennas under bending. **f** Photograph shows that the electrical energy transmitted to MXene NFC antenna from an NFC-enabled smartphone is able to light up 168 parallel LED lights. **g** Optical images of two small-sized circular (top) and square (bottom) MXene NFC tags with the lighted LEDs. **h** Two application examples of the MXene NFC tags, access card for standard electronic door locks (left) and identification label for plant information (right).

the extruded fillaments to maintain the solid shape without spreading. When using smaller needles, the printed line width can be further reduced to 120 μm (Supplementary Fig. 12). The almost identical Raman spectra of two-year-old lines supported on different low-cost, popular substrates (Fig. 2g and Supplementary Figs. 13 and 14) suggests that (1) pristine MXenes have well preserved along/after the extrusion, and (2) the direct printing strategy of MXene-based electronics is universal and compatible with existing thin-film technologies.

We then characterized the MXene printed films. Due to the shear-induced alignment during extrusion, the as-printed paths are composed of densely stacked and interconnected $Ti_3C_2T_x$ nanosheets (Fig. 2h and Supplementary Fig. 15), forming a robust metallic network that allows free and fast electron transport (dominated by the intrinsic intra-flake processes)[26], thereby enabling high conductivity and mechanical flexibility (as evidenced by the cyclic bending test, Supplementary Fig. 16). Figure 2i suggests that increasing <N> results in thicker films with lower sheet resistance. Notably, the printed thickness scales linearly with <N>, another indicative of high-resolution prints with sharp printing edges (Supplementary Fig. 17); otherwise, the thickness deviates from the fitted line because of the dome formation. Based on the sheet resistance and thickness, the electrical conductivity of all-printed lines was obtained, showing a value up to 6260 S cm⁻¹ when <N> = 2 right after printing (Fig. 2j), which can be further improved to 6900 S cm⁻¹ by simply storing in low-humidity condition (~10% RH) for 4 h. We note the direct printing of MXene inks at room temperature to achieve metallic conductivity possesses apparent advantages over the

printing of liquid metals or other metal-based inks (such as Zn, Ag nanoparticles, Supplementary Table 2), as the latter require either UV curing or annealing, which is not plausible for printed electronics mounted on temperature-sensitive, low-cost substrates.

Considering that high concentration (c) and electrical conductivity (σ) are both indispensable for printable inks to achieve high-efficiency printing, we use a key figure of merit (FoM = σc) to evaluate the printing efficiency as previously recommended[8,27]. A higher FoM means a faster printing speed with a higher conductivity in a given film thickness. As shown in Fig. 2k, the formulated MXene ink reaches a record-high FoM of ~414,000 S cm⁻¹ mg mL⁻¹ (<N> = 2), exceeding that of all other reported printable inks[27]. The preferable rheological, electrical, and mechanical properties of MXene inks suggest the great plausibility of room-temperature printing of high-performance flexible wireless electronics, as discussed below.

**All-MXene-printed NFC devices.** In the era of IoTs, high-performance integrated antennas are indispensable for flexible, portable electronics[28] due to their capability in wireless data transmission and energy harvesting. Near-field communication (NFC) is a short-range wireless technology that allows simultaneous power and data transmission between devices through inductive coupling, offering a versatile platform for battery-free miniature sensing electronics[29]. As such, we fabricated the first all-MXene-printed NFC antenna at room temperature with 70 mm in length and 45 mm in width, referencing the standard size of credit cards (Fig. 3a, b and Supplementary Fig. 18).

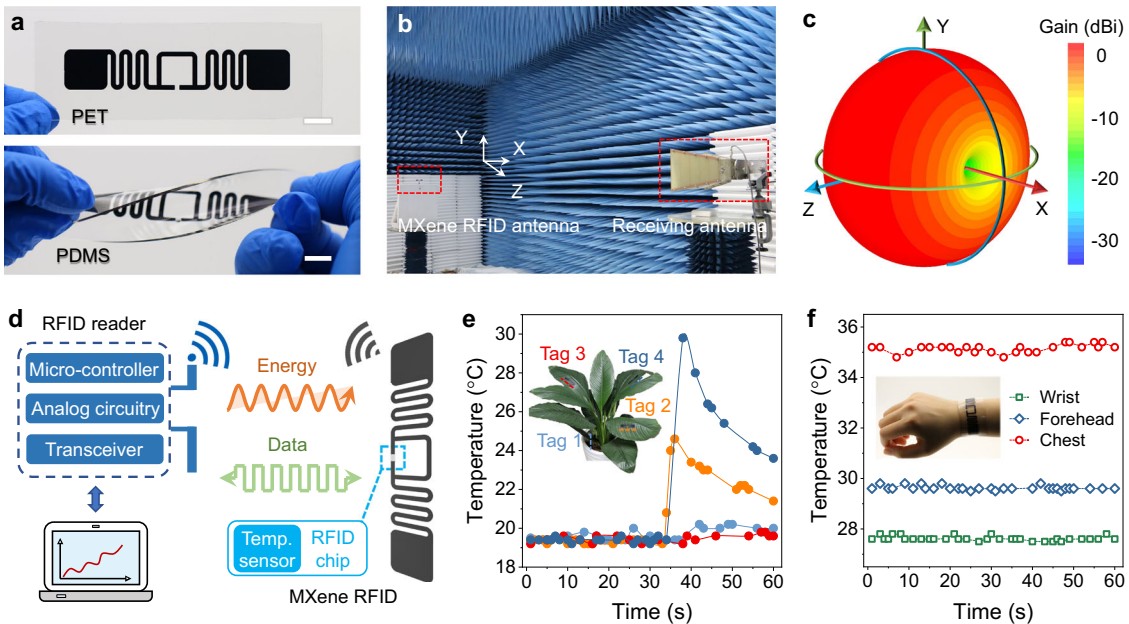

**Fig. 4 Demonstration of the MXene RFID temperature sensing system. a** Optical images of the MXene dipole antennas with a size of 15 mm × 86 mm on PET (top) and PDMS (bottom). Scale bar, 10 mm. **b** Antenna radiation pattern measurement in the anechoic chamber. **c** Simulated 3D radiation pattern of the MXene dipole antenna at 920 MHz. **d** Schematic mechanism of the MXene RFID temperature tag with a laptop-connected RFID reader. **e** Local monitoring of leaf surface temperature using four MXene RFID temperature tags mounted on the leaf surface and plant root (inset). **f** MXene RFID temperature tags as wearable sensors to monitor surface temperature on the human wristband (inset), forehead, and chest.

To maximize the quality factor $Q$, the coil geometry was pre-simulated, as summarized in Supplementary Table 3. As a result, the number of turns of 5 coils, line width of 2.5 mm, and coil spacing of 0.5 mm were selected as the specific geometrical parameters, covering the communication needs of most commercial NFC-enabled devices[30]. Figure 3c presents the antenna surface current distribution at 13.56 MHz. When the thickness of MXene antenna is below the skin depth, the antenna resistance and $Q$ factor are adjustable (Fig. 3d). The MXene antenna coils can be printed on different functional substrates, including biodegradable poly(vinyl alcohol) (PVA), ferrite substrates that shield the antenna from metal interference, and other nine different types of substrates, with excellent mechanical robustness (Supplementary Figs. 19 and 20 and Movie 2). The encapsulated MXene NFC tags have a lifespan comparable to the commercial tags. Interestingly, for the MXene NFC tag without encapsulation, it still works properly after two years of low-humidity storage (Supplementary Fig. 21), attributed to the as-formed surface oxide layer protecting the internal conductive path[31]. Furthermore, flexible MXene antenna coils with personalized smaller sizes were also fabricated (Supplementary Fig. 22a–d). The finite element analysis (FEA) results reveal that the strain of the MXene antenna is mainly distributed in the axial direction of bending during deformation (Fig. 3e). Throughout repeated bending (Supplementary Fig. 22e–l), the MXene tag maintains its integrated structure without any cracks, indicating the sandwich structure protects the MXene interlayer between PDMS films from reaching the yield strain.

The as-printed MXene NFC antennas showcase stable accessibility at 13.56 MHz for various NFC-enabled equipment. The wirelessly harvested energy from a smartphone *via* the robust MXene NFC antenna is able to light hundreds of LEDs (Fig. 3f and Supplementary Movie 3), indicative of excellent energy transmission capability. The combination of data exchange property suggests their great potential to replace these existing commercial ones and find applications in the next-generation, battery-free, wireless sensing devices, such as critical information

recognition and energy harvesting (Fig. 3g), wearable electronic ID cards, and flexible plant identification labels (Fig. 3h and Supplementary Movie 4), to name just a few.

**All-MXene-printed RFID sensing system**. As another core wireless IoT technology, radio frequency identification (RFID) is promising for long-range wireless sensing due to its unique advantages of non-contact, low power consumption, rapid identification, etc.[32]. As such, we demonstrated the first printed flexible RFID passive tags for wireless temperature sensing. The folded dipole with a closed-loop was chosen as the RFID antenna design, offering great freedom for geometry optimization and chip impedance matching[33]. Figure 4a presents the MXene dipole antennas on PET and PDMS, revealing high printing precision and excellent mechanical flexibility. To evaluate the antenna radiation properties, the far-field radiation pattern was measured in an anechoic chamber (Fig. 4b). The MXene dipole antenna exhibits a desirable omnidirectional radiation pattern in the H plane and a dipole-shaped pattern with symmetrical angle-dependent gain values in the E plane, as evidenced by the simulated and measured results (Fig. 4c and Supplementary Fig. 23a). The dipole antenna's resonant frequency and reflection $S_{11}$ are presented in Supplementary Fig. 23b, exhibiting a wide frequency band with a peak near 920 MHz even after 500 bends. The high current density in the middle section of the antenna is sufficient to satisfy the microchip power demands (Supplementary Fig. 24).

The working mechanism of the MXene RFID temperature monitoring system is illustrated in Fig. 4d, which is based on the backscatter coupling between the reader and the RFID tag[34]. The entire RFID reading system is simple and easy to operate (Supplementary Fig. 25). As a demonstration, we use this system to detect the local temperature on plants and human body, exhibiting high consistency and sensitivity upon temperature variations (Fig. 4e, f and Supplementary Movie 5). Considering the current enormous quantity of metal tags and the e-waste risk,

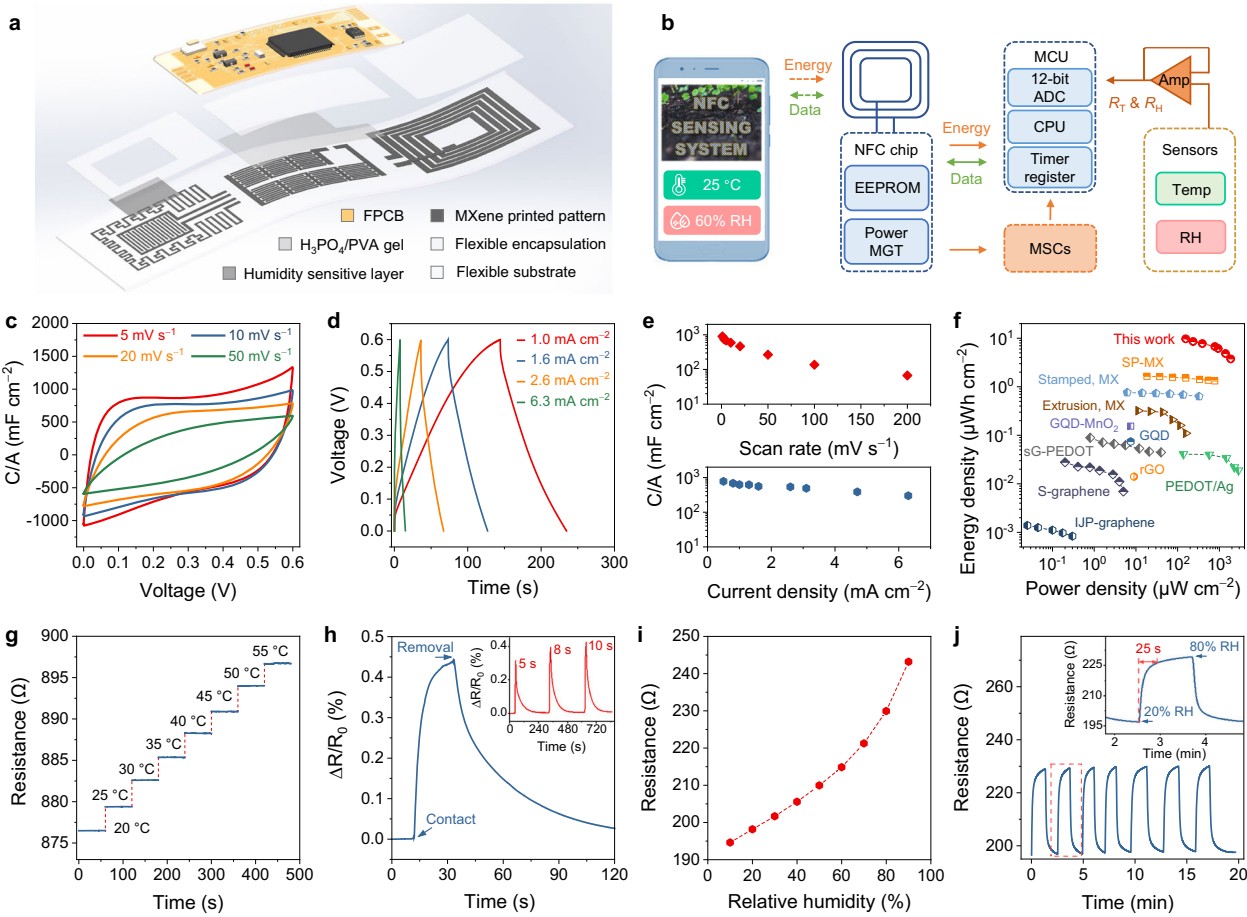

**Fig. 5 Demonstration of flexible printed integrated wireless sensing electronics. a** Schematic illustration of the wireless sensing system integrated with three all-MXene-printed functional modules. **b** Block diagram and operating principle of the integrated sensing system. **c–e** CV and GCD curves of the MSC unit at different scan rates (**c**) and current densities (**d**) with areal capacitance (C/A) calculated from CV (top) and GCD (bottom) shown in **e**. **f** Ragone plots present the comparison of areal energy and power density of this work with other reported MSC systems. More comparison information is presented in Supplementary Table 6. **g** Dynamic electrical responses of the MXene temperature sensor in the range of 20–55 °C. **h** Normalized resistance responses upon contact and removal from the palm. Inset: the electrical response changes under different contact time. **i** Electrical response changes of the MXene humidity sensor under various humidity conditions. **j** Cyclic dynamic electrical response and recovery curves measured from 20% RH to 80% RH. Inset: the response time.

our eco-friendly, low-cost MXene RFID represents a sustainable alternative for radio frequency electronics[35].

**All-MXene-printed flexible wireless integrated sensing system.** Based on the excellent wireless sensing and outstanding power/ data transmission properties, we further directly printed the abovementioned MXene components on PDMS at room temperature and integrated them with a flexible printed circuit board (FPCB), forming a flexible integrated system (Fig. 5a and Supplementary Figs. 26 and 27). The FPCB is a control module connected to three printed modules (Supplementary Fig. 28 and Table 4), thus enabling wireless communication with smartphones for temperature and humidity (T/H) sensing and energy transmission. The as-transmitted energy can also be simultaneously stored in the MSC module to power the sensors once the smartphone is removed. The working principle of the integrated wireless sensing system is demonstrated in Fig. 5b, showing the power delivery and data acquisition pathways.

As such, we evaluated the energy storage properties of the directly room-temperature printed MXene MSC module. A single MSC device (interdigitated finger gap ~200 μm, Supplementary Fig. 29a), with PVA/H₃PO₄ gel as the semi-solid electrolyte, exhibits a capacitive charge-storage behavior, as seen by the quasi-rectangular cyclic voltammogram (CV) shape and quasi-

linear galvanostatic charging-discharging (GCD) curves at different rates (Fig. 5c, d). The areal capacitances obtained via CV and GCD are comparable (Fig. 5e), reaching up to ~900 mF cm⁻², which has exceeded those of previous reports on printed planar MSCs (Supplementary Fig. 29b and Table 5)[12]. Figure 5f compares the Ragone plots of various MSC devices. The calculated areal energy density and power density of our MXene MSC are up to 9.7 μW h cm⁻² and 1.875 mW cm⁻², respectively, being orders of magnitude higher than those of planar MSCs (Supplementary Table 6). By designing the configuration (series/ parallel) of extrusion-printed MXene MSC units, one can realize a module satisfying different power/energy demands. Electrochemical characterizations further confirm that rapid charge transfer kinetics and satisfying cycling (~90% after 3000 cycles) were achieved in the 7-printed MXene MSC module within 3 V (Supplementary Fig. 29c–g), which is able to power the microgrid and systems[36], such as LEDs or the like (Supplementary Fig. 29h).

We then explored the sensing performance of this integrated system (Supplementary Figs. 30 and 31). Due to its large thermal expansion coefficient, PDMS substrate tends to expand as the temperature rises, resulting in the stretching of the MXene conductive network and varying the corresponding electrical conductivity[37]. As shown in Fig. 5g, our MXene temperature sensor exhibits a positive temperature coefficient behavior

(sensitivity of ~0.066% °C$^{-1}$), with the ability to respond rapidly to temperature changes (Fig. 5h and Supplementary Fig. 32). For the humidity sensor, a thin MXene film was utilized as the humidity sensing layer (Supplementary Fig. 33), as MXenes display a sharp electrical variation upon the widening of nanosheets' interlayer spacing when exposed to the humid condition[38] (Fig. 5i). During the cyclic humidity test, a repeatable and stable resistance variation (response time of ~25 s) was observed (Fig. 5j). As a functional demonstration, Supplementary Movie 6 shows the operation process of the integrated sensing system for wireless monitoring of plant T/H microenvironment. The quick response may suggest the promising application of all-MXene-printed integrated systems in smart sensing agriculture.

## Discussions

This is the first attempt to fabricate all-MXene-printed wireless sensing electronics at room temperature, with promising preliminary performance in energy/data transmission/sensing behaviors coupled with excellent mechanical robustness. We firmly believe there is still plenty of room for performance enhancement. In terms of the MSCs, diverse strategies, such as optimizing printed gap and thickness, adjusting surface chemistries, and utilizing asymmetric configurations, are promising routes to further improve their charge storage property and energy/power densities[8,12,22]. As for the sensing functions, the existing T/H module can be further optimized through structural design or nanomaterials modification[39]. Considering the diverse MXene family consisted of >30 versatile members and is still quickly expanding, more advanced MXene-based wireless electronics may be enabled by either choosing novel MXene inks and/ or the booming printing/wireless technologies[40,41] or varying the energy storage devices (such as flexible batteries, solar cells, TENGs, etc.)[42]/sensing modules (such as flexible chemical, physical, and biological sensors)[43,44] etc.

To summarize, we have reported the room-temperature high-precision printing of flexible wireless electronics using additive-free MXene aqueous ink for the first time. The desirable ink rheological and electrical properties make our MXene inks extremely suitable for high-precision extrusion printing on different plane/curved substrates without annealing, exhibiting metallic conductivity (up to 6900 S cm$^{-1}$), ultranarrow printing line gap (3 μm), and high spatial uniformity (within 0.43%) in the printed tracks. In particular, the ultranarrow printed line gap represent the state-of-the-art direct printing using nanomaterials at room temperature, representing the high-efficient printing of high packing density electronics. Using this approach, distinct high-performance modules can be easily fabricated solely or integrating with other electrical components, holding a great promise in replacing some cumbersome e-waste electronics such as commercial antennas.

As a conceptual exhibition, we demonstrated an all-MXene-printed monolithic flexible integrated system that enables simultaneous wireless power harvesting, data transmission, and T/H sensing. This room-temperature printing strategy shows enormous application potential for flexible electronics fabrication in diverse fields like IoTs, smart labels, intelligent packaging, environment monitoring, agriculture sensing, healthcare, 5G, etc. Looking ahead, we expect this work will inspire more exploration of easy-to-integrate printable components and accelerate the progress of printed flexible, wearable electronics.

## Data availability

The datasets generated during and/or analyzed during the current study are available from the corresponding author on reasonable request. Source data are provided with this paper.

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

## Acknowledgements

This research was supported by the National Science Fund for Excellent Young Scholars of China (Grant No. 31922063) and the Fundamental Research Funds for the Central Universities. We thank the supports from ETH Board, and the Empa internal research call grants (IRC-cupsupercap 2019, IRC-NitfixMX 2020). We thank X. Li, G. Xu, and Q. Liu for helpful discussions, and D. Li, H. Dong, and J. Xie for assistance in device fabrication and measurement. We thank the technical support from the State Key Laboratory of Modern Optical Instruments and the Analysis Center of Agrobiology and Environmental Sciences, Zhejiang University.

## Author contributions

Y.S., Y.Y., J.P., and C.Z. designed the project and experiments. Y.S., C.Z., C.J., and Y.Yao performed materials synthesis, ink development, and characterization experiments. Y.S., C.Z., B.P., and H.C. fabricated the printed devices, carried out the measurements, and analyzed the data. S.W. and J.H. helped with the RF design and software simulations. X.W. performed the circuits design and measurements. Y.S., J.P., and C.Z. wrote the manuscript. All the authors reviewed and commented on the manuscript.

## Competing interests

The authors declare no competing interests.
