## [Peer Review File · Nature Communications]

REVIEWER COMMENTS

Reviewer #1 (Remarks to the Author):

Ping et. al. demonstrated the flexible wire-less electronic device with room-temperature printing of MXene inks. Authors optimized the strategy for fabricating a stable and high dispersed MXene ink without any treatments or additives, thereby leading to good electrical conductivity and rheological property suitable to the extrusion printing. In addition, the resulting MXene ink allowed high solid concentration up to 60mg/mL, as well as showed good stability for up to two years. I think this ink can give significant and practicably assistance to the printed electronic field. So, I recommend this article to be published after revising some issues.

1. Authors fabricated several printed electronic devices with only extrusion printing method. However, several different printings (e.g., inkjet, slot-die coating and so on) have been frequently applied to make electronic device. So, authors should demonstrate the universality of your MXene ink to apply it to various printing method.

2. Authors reported that the formulated MXene ink reaches a record-high FoM of $\sim 414,000 \text{ S cm}^{-1} \text{ mg mL}^{-1}$ ($\langle N \rangle = 2$). However, the relevant discussion is too short and a bit lacking. So, Authors should discuss the relationship between electrical conductivity, rheological properties and MXene morphology in detail.

3. The printed MXene patterns showed good electrical performance and good pattern fidelity. In particular, the gap between patterns reaches down to 3 μm , which is very nice result. However, the minimum width of the pattern is about 220 μm , which is not good. So, it is necessary to reduce the width of pattern to apply your ink to the microelectronics field.

4. The optimum height of the patten is about 750 nm for the sample with $\langle N \rangle = 2$. If the MXene ink is applied to the TFT or other practical applications, the height of the pattern needs to be less than 100 nm. So, it would be instructutive to discuss the electrical conductivity of the smaller pattens (width and height of the pattern are less than 100 μm and 100 nm, respectively)

5. The electrical conductivity of the MXene pattern dependent on the "humidity" should be discussed. In addition, Authors need to show storage stability of the ink under the high humidity and temperature condition.

Reviewer #2 (Remarks to the Author):

This is an interesting and timely article of potential scientific and technological impact. Due to their high electronic conductivity and versatility, MXenes are of great potential interest for printed, wearable and flexible electronics, which is a hot research topic now. MXenes by far outperform graphene and other carbon nanomaterials in those applications. Metallic conductivity and hydrophilicity of $\text{Ti}_3\text{C}_2\text{T}_x$ MXene is particularly attractive for printing highly

conductive patterns. Inkjet and screen printing with additive-free MXene inks have been reported, but only a couple of studies on extrusion printing of MXenes have been published. The videos accompanying the paper are impressive and clearly demonstrate the potential of the method. I can recommend this paper for publication after minor revision.

Below are specific comments to be addressed.

1. The authors should provide a better comparison and discussion of ink-jet and extrusion printing in terms of the applications. For example, comparable values of conductivity have been achieved by inkjet printing (see Ref. 19 in SI).
2. The use of additive-free MXene inks for printed electronics was first proposed in E. Quain, et al, *Direct Writing of Additive-Free MXene-in-Water Ink for Electronics and Energy Storage, Advanced Materials Technologies*, 4 (1) 1800256 (2019), not in Ref. 16.
3. The authors claim stability of the printed patterns. Have they measured the conductivity of the printed patterns as a function of time? Water adsorption and swelling of the films may affect the electrical characteristics of MXenes.
4. Schematic 1 in SI implies the orientation of MXene flakes parallel to the substrate surface. Is this really the case? Does it depend on the ink viscosity and printing parameters? The authors should provide information about orientation of flakes and SEM images of the printed line cross-section and surface, as flakes orientation will affect electronic transport and other properties.
5. The authors demonstrate printing on a variety of substrates. How does ink spreading (bleeding) depend on the substrate?
6. Very recently (already after submission of the current manuscript), a related paper was published in *Nature Electronics* by V. Nicolosi and F. Torrioni groups: E. Piatti et al., *Charge transport mechanisms in inkjet-printed thin-film transistors based on two-dimensional materials, Nature Electronics*, 4, 893–905 (2021). It discusses the mechanism of charge transfer in printed MXene films and may help with the explanation of the results reported in this work.
7. The authors should minimize the use of superlatives, such as "ideal", etc. There is no ideal ink – there is always room for further improvement.

We acknowledge the reviewers for their time and very helpful comments. Provided below is our detailed response to each comment/suggestion. The specific changes made to the manuscript to address each point are highlighted in yellow.

Responses to Reviewers' Comments:

Reviewer #1

Ping et. al. demonstrated the flexible wire-less electronic device with room-temperature printing of MXene inks. Authors optimized the strategy for fabricating a stable and high dispersed MXene ink without any treatments or additives, thereby leading to good electrical conductivity and rheological property suitable to the extrusion printing. In addition, the resulting MXene ink allowed high solid concentration up to 60mg/mL, as well as showed good stability for up to two years. I think this ink can give significant and practicably assistance to the printed electronic field. So, I recommend this article to be published after revising some issues.

Response: We would first like to thank the reviewer for the positive comment on the significance and quality of our work. Your insightful comments are very constructive in further improving our work. We have tried our best to revise our manuscript accordingly. Below are addressed all the comments point by point.

1. Authors fabricated several printed electronic devices with only extrusion printing method. However, several different printings (e.g., inkjet, slot-die coating and so on) have been frequently applied to make electronic device. So, authors should demonstrate the universality of your MXene ink to apply it to various printing method.

Response: We would like to thank the reviewer for this valuable comment about the universality of our developed MXene inks, which is a very important issue concerning printing materials and technologies. MXene aqueous inks have a large concentration window, so they can

be specifically configured according to the viscosity requirements of different printing methods (such as changing the concentration, flake size, monolayer ratio, etc.). With years of continuous research work, we have rich experience in the preparation and application of MXene inks. Before this work, we have systematically studied and summarized the properties and applications of MXenes (e.g., *Adv. Funct. Mater.* 26, 4143-4151 (2016); *Chem. Mater.* 29, 4848-4856 (2017); *Mater. Today* 48, 214-240 (2021).)¹⁻³, and demonstrated the great potential of MXene ink for stamp printing (*Adv. Funct. Mater.* 28, 1705506 (2018))⁴, inkjet printing (*Nat. Commun.* 10, 1795 (2019))⁵, and screen printing (*Adv. Mater.* 32, e2000716 (2020))⁶. In brief, we demonstrated various printing strategies using MXene inks with different ink rheological properties that were specifically configured. In this work, extrusion printing was chosen based on its excellent application value in the fabrication of flexible electronics, and this work further demonstrates the great universality of MXene inks. To better illustrate this point, we highlight in our revised manuscript that our MXene aqueous inks are extrusion printable while also further emphasizing the remarkable printing universality of MXene:

"MXene aqueous inks have a large concentration window, allowing the rheological properties to be adjusted according to the printing method." (Supplementary Fig. 1)

"The concentration of $Ti_3C_2T_x$ ink was controlled at $\sim 60 \text{ mg mL}^{-1}$ by the addition of water to achieve appropriate rheological properties for direct extrusion printing." (Supplementary Notes)

2. Authors reported that the formulated MXene ink reaches a record-high FoM of $\sim 414,000 \text{ S cm}^{-1} \text{ mg mL}^{-1}$ ($\langle N \rangle = 2$). However, the relevant discussion is too short and a bit lacking. So, Authors should discuss the relationship between electrical conductivity, rheological properties and MXene morphology in detail.

Response: Many thanks for this constructive suggestion. In our work, we use a key figure of merit (FoM = σc) to evaluate the printing efficiency as previously recommended (*Nat. Commun.*

10, 1795 (2019))⁵, which is based on the consideration that high concentration (c) and electrical conductivity (σ) are both indispensable for printable inks to achieve high-efficiency printing. The record-high FoM represents that our developed MXene inks can accomplish a faster printing speed with a higher conductivity in a given film thickness. The fundamentals for achieving this are the MXene's own characteristics, that is, its remarkable conductivity and a large concentration range for aqueous inks. In this work, we first introduce the preparation and features of our developed MXene ink before introducing the concept of FoM and characterize the parameters of direct MXene printing in detail, especially focusing on the printing conductivity. In addition, we provide a more detailed description and explanation about this section in Supplementary Materials. For example, Supplementary Fig. 1 shows the schematic diagram of preparation and extrusion printing of $\text{Ti}_3\text{C}_2\text{T}_x$ MXene ink. Supplementary Fig. 2 demonstrated the large single-layer ratio (>90%) and narrow flake size distribution of our developed MXene inks. The higher portion of predominantly single-layer, larger flakes in our inks suggests much higher viscosity, storage modulus (G'), and loss modulus (G'') compared to those of traditional inks, all of which are highly preferred in the extrusion printing for fine lines or structures. Moreover, these improved rheological properties are expected to enhance the electrical properties of the MXene ink, boost the extrusion printing precision, and prevent oversized nanosheets from accumulating and clogging the nozzle during extrusion printing. And we further discuss the importance of suitable rheological properties to achieve high-quality printing in Supplementary Fig. 3. Below are the corresponding changes we made in the revised manuscript:

"Due to the shear-induced alignment during extrusion, the as-printed paths are composed of densely stacked and interconnected $\text{Ti}_3\text{C}_2\text{T}_x$ nanosheets (Fig. 2h and Supplementary Fig. 3), forming a robust metallic network that allows free and fast electron transport (dominated by the intrinsic intra-flake processes), thereby enabling high conductivity and mechanical flexibility (as evidenced by the cyclic bending test, Supplementary Figs. 12)." (Page 6 in manuscript)

Supplementary Fig. 1 | Schematic diagram showing the preparation and extrusion printing of $\text{Ti}_3\text{C}_2\text{T}_x$ MXene ink. MXenes are a new 2D family of transition metal carbides and nitrides derived from the etched MAX precursor, with a general formula of $\text{M}_{n+1}\text{X}_n\text{T}_x$ ($n = 1-4$), where M stands for the transition metal, X denotes carbon or nitrogen, and T is the surface termination groups (e.g., -O, -OH, or -F). MXene aqueous inks have a large concentration range, allowing the rheological properties to be adjusted according to the printing method. Here, concentrated $\text{Ti}_3\text{C}_2\text{T}_x$ MXene aqueous inks were prepared *via* the modified MILD synthesis route, followed by optimized centrifugation and sonication, such that a high percent of single-layer flakes (>90%) with a narrow sheet diameter distribution are enabled. Consequently, the rheological properties of the as-formulated MXene inks fulfill the requirements of the extrusion printing very well, leading to high printing resolution and precision. Due to the shear force created in the nozzle of the printing head, $\text{Ti}_3\text{C}_2\text{T}_x$ nanosheets align themselves in the axial direction of the print head, forming highly-ordered liquid crystals within the extruded filaments and resulting in MXene films with densely stacked flakes in parallel to the substrate. Consequently, a high metallic electrical conductivity is enabled in the extrusion-printed MXene tracks. This result was also confirmed in subsequent experiments on the printed MXene films, such as XRD, SEM, and electrical measurements.

Supplementary Fig. 15 | Microscopic surface morphology of the printed MXene film. a, A cross-sectional SEM view of the printed MXene film on PET. Scale bar, 20 μm . **b,** SEM image shows densely stacked MXene flakes on the surface of the printed MXene film. Scale bar, 500 nm. The cross-section and top view demonstrate the continuous coverage of the printed MXene film on the substrate without protruding flake corners. Due to the shear force during extrusion printing and the 2D material properties of MXene, the $\text{Ti}_3\text{C}_2\text{T}_x$ flakes prefer alignment parallel to the substrate after printing, resulting in dense and highly conductive films. According to recent research, the fast charge transport within the printed MXene films is dominated by the intrinsic intra-flake processes.

3. The printed MXene patterns showed good electrical performance and good pattern fidelity. In particular, the gap between patterns reaches down to 3 μm , which is very nice result. However, the minimum width of the pattern is about 220 μm , which is not good. So, it is necessary to reduce the width of pattern to apply your ink to the microelectronics field.

Response: We would like to thank the reviewer for this insightful suggestion. Resolution is one of the critical indicators of printing technology, and the resolution of extrusion printing mainly depends on the size of the extrusion needle. In this work, our printing goal is to use additive-free MXene ink to achieve the printed fabrication of antennas, micro-supercapacitors, and sensors. The size of these devices is usually in the centimeter level, and the precision requirements are also not as harsh as the micrometer level. Therefore, our current printing parameters are entirely

in line with these requirements; that is, a minimum line width of 220 μm and a minimum spacing of 3 μm are achieved with a 110- μm extrusion nozzle. But as for the field of microelectronics, such a line width may not be enough, which is also a problem we have considered. The most straightforward way to solve this problem is to choose a nozzle with a smaller diameter. However, the accompanying problem is that the reduction of the needle size also puts forward higher requirements for configuring the extrusion printable ink, involving concentration, viscosity, sheet diameter, etc. It has been estimated that particles should be kept below 1/100 of the capillary radius to prevent clogging, placing a strong requirement for well-dispersed MXene ink systems in high-resolution printing (*Adv. Funct. Mater.* **22**, 4790-4800 (2012))⁷. Therefore, high-precision printing with a line width of fewer than 100 μm is not an easy task for MXene inks with large flake diameters. In this regard, we also tried to do some preliminary exploration. Using a custom-made needle (inner diameter 50 μm) and reconfigured MXene ink (15 mg mL^{-1}), we demonstrate that the printed line width can be further reduced to 120 μm using extrusion printing (as shown in Supplementary Fig. 2). This printing parametric parameter is close to a recently reported work that fabricated MXene ink-based TFT using electrohydrodynamic (EHD) printing (*Adv. Funct. Mater.* **31**, 2010897 (2021))⁸. We believe a narrower printed line width can be achieved by continuing to reduce the diameter of the needle, *but due to the pandemic, we are currently unable to purchase smaller custom-made needles*. If conditions permit, we will continue to dig deeper into MXene-printed microelectronics.

4. The optimum height of the patten is about 750 nm for the sample with $\langle N \rangle = 2$. If the MXene ink is applied to the TFT or other practical applications, the height of the pattern needs to be less than 100 nm. So, it would be instructutive to discuss the electrical conductivity of the smaller pattens (width and height of the pattern are less than 100 μm and 100 nm, respectively).

Response: We would like to thank the reviewer for this thoughtful recommendation. For the preparation of printed microelectronics, the selected printing method must meet the parameter

requirements of the device. Based on this essential requirement, if we want to expand additive-free inks into microelectronics, we need to find smaller needles and develop inks with corresponding rheological properties. In addition, we also need to consider factors such as solvent selection and evaporation kinetics, MXene flake size, monolayer ratio, and so on; because many effects will be amplified after printing into the microscale. Since our work mainly revolves around applying additive-free MXene aqueous ink in printed flexible wireless electronics, we focus more on high-throughput printing of functional inks. Therefore, we would only make some preliminary attempts here for microelectronics applications. We demonstrate that the printed MXene line widths can be further reduced to 120 μm when using custom needles (inner diameter 50 μm) and reconfigured MXene ink (15 mg mL^{-1}). Measured by the white light interferometer, the thickness of these lines is approximately 20 nm. Electrical measurements proved these lines are continuous and conductive. These printing parameters are sufficient for many microelectronics. Recently, we tried to use this printing technology to develop organic electrochemical transistors (OECTs), and the relevant content will be reported in detail in subsequent work. Accordingly, we have also made corresponding changes to this part in the revised manuscript:

Supplementary Fig. 3 | Further reduction of printed line width. a, SEM image of the printed MXene lines on PET ($\langle N \rangle = 1$). Scale bar, 200 μm . **b**, Height profiles of the printed MXene lines on PET. **c**, Current-voltage plots of the printed MXene lines with different lengths. Considering

the potential applications in microelectronics, we try to reduce the printed line width. We demonstrate that the printed MXene line widths can be further reduced to 120 μm with an average height of ~ 20 nm when using custom needles (inner diameter 50 μm) and reconfigured MXene inks (15 mg mL⁻¹). Further, electrical measurements prove these lines are continuous and conductive. The size of the extrusion needle dominates the width of the printed line, and narrower print line widths can be achieved by using smaller needles. However, when using smaller needles, it is also necessary to configure inks with suitable rheological properties, including control of flake size to prevent clogging.

5. The electrical conductivity of the MXene pattern dependent on the "humidity" should be discussed. In addition, Authors need to show storage stability of the ink under the high humidity and temperature condition.

Response: We would like to thank the reviewer for this constructive comment. The metallic conductivity of MXenes is a well-known distinguishing feature compared to other 2D nanomaterials. Of course, the electrical conductivity of MXene has been proved to be humidity dependent, which is because of the effect of water molecules on the interlayer spacing between nanosheets. When the humidity increases, the electrical conductivity of the MXene films decreases due to the increase in the flake interlayer spacing. On the other hand, low-humidity treatment of the printed MXene film can also significantly improve its electrical conductivity. This alternative to high-temperature annealing provides a novel avenue for printing conductive circuits on flexible substrate materials that cannot withstand high temperatures. Based on this principle, many reports have been developing humidity sensors using MXenes as humidity-sensitive materials. In this work, our developed MXene-based humidity sensors present a continuous increase in electrical resistance for increased RH from 20% to 80%. However, the humidity-dependent effect of the electrical conductivity decreases as the thickness of the MXene

film increases. More details on the humidity dependence of electrical conductivity can be found in the section of the all-MXene-printed wireless flexible integrated sensing system in our article.

Regarding the stability of MXene ink, we have also paid great attention to this concern during the experiment process and described it in the article. Our developed MXene aqueous ink shows good stability without sedimentation when stored in Ar-sealed bottles in the dark and low temperature (<4 °C) for at least two years, ensuring a sufficient time window for potential ink processing. Considering this issue raised by the reviewers, we also believe that stability testing of the ink in a non-refrigerator environment is necessary. Regarding the storage of the MXene ink, we tested its stability under ambient conditions, which is the most common and convenient storage condition. According to our previous research experience on MXene stability (*Chem. Mater.* 29, 4848-4856 (2017))², the coagulation and oxidation of MXene nanosheets are significant factors affecting the storage of MXene colloidal solutions. In the experiments, we used nitrogen to exclude the dissolved oxygen in the water to avoid the oxidation of the nanosheets. Otherwise, the MXene nanosheets would be oxidized, leading to the formation of cloudy-white colloidal solutions containing primarily anatase (TiO₂). Besides, our developed MXene inks have a concentration of up to 60 mg mL⁻¹, and the high solid content can avoid the problem of nanosheets coagulation. As a result, the developed MXene inks still have good stability under ambient conditions. In the revised manuscript, we further detail our conductivity testing conditions and provide the results of the stability test of the MXene inks under ambient conditions. The following are the corresponding changes made in the revised manuscript:

"The measurement was carried out after the initial printing of MXene film at ambient conditions (25 °C, 40% RH) for 10 min and further drying in low humidity conditions (<10% RH) for another 4 hours, respectively." (Supplementary Notes)

Supplementary Fig. 4 | Long-term testing of MXene aqueous ink stability under ambient conditions. a-f, Seven inks with concentrations (left to right: 60 mg mL^{-1} , 40 mg mL^{-1} , 20 mg mL^{-1} , 10 mg mL^{-1} , 5 mg mL^{-1} , 1 mg mL^{-1} , 0.2 mg mL^{-1}) were configured, and the stability was observed for one month. Here, N_2 was used to eliminate dissolved oxygen in the water to avoid the oxidation of the nanosheets. During a month of observation, the most obvious phenomenon is the aggregation and re-stacking of MXene flakes due to van der Waals forces in low concentration inks. However, this phenomenon does not occur in high concentration inks because of the high solid content. In addition, no apparent oxidation phenomenon is found in these inks with different concentrations, that is, no the formation of cloudy-white colloidal solutions containing primarily anatase (TiO_2). In conclusion, the developed MXene aqueous inks still have good stability under ambient conditions.

Finally, we would like to thank the reviewer again for these valuable comments and for the thoughtful and careful review towards improving our manuscript.

Reviewer #2

This is an interesting and timely article of potential scientific and technological impact. Due to their high electronic conductivity and versatility, MXenes are of great potential interest for printed, wearable and flexible electronics, which is a hot research topic now. MXenes by far outperform graphene and other carbon nanomaterials in those applications. Metallic conductivity and hydrophilicity of $\text{Ti}_3\text{C}_2\text{T}_x$ MXene is particularly attractive for printing highly conductive patterns. Inkjet and screen printing with additive-free MXene inks have been reported, but only a couple of studies on extrusion printing of MXenes have been published. The videos accompanying the paper are impressive and clearly demonstrate the potential of the method. I can recommend this paper for publication after minor revision.

Response: Many Thanks. We are sincerely grateful for your positive comments on our work. Your suggestions are professional and constructive, and we have tried our best to revise our manuscript accordingly. Below are addressed all the comments point by point.

Below are specific comments to be addressed.

1. The authors should provide a better comparison and discussion of ink-jet and extrusion printing in terms of the applications. For example, comparable values of conductivity have been achieved by inkjet printing (see Ref. 19 in SI).

Response: We would like to thank the reviewer for this valuable comment. Ref. 19 in SI (*Small* 17, 2006376 (2020))⁹ is an excellent work that reported the thermal inkjet printing of additive-free aqueous MXene inks on textiles. The author focuses on increasing the functional capacity of conductive inks and simplifying the fabrication of wearable textile-based electronics. The reason for choosing thermal inkjet (TIJ) printing is because of its potential to rapidly produce textile devices that are flexible, complex, and have relatively large circuits for use in numerous applications. However, the advantages of extrusion printing over inkjet printing can be

more clearly demonstrated when it comes to manufacturing flexible wireless electronics. As pointed out in our previous "property-process map" summary for MXene processing (*Mater. Today* 48, 214-240 (2021))³, extrusion printing has apparent advantages over inkjet printing in ink viscosity, printing throughput, and film thickness. Given the wider range of ink viscosity, extrusion printable inks tend to have larger FoM values ($\text{FoM} = \sigma c$) than inkjet printable inks. In our work, a record-high FoM value was achieved, which represents that our developed MXene inks can achieve a faster printing speed with a higher conductivity in a given film thickness. In other words, even at similar ink conductivities, it is easier to produce highly conductive printed patterns using high-throughput extrusion printing, which is essential for some applications (such as wireless communications). On the other hand, compared with inkjet printing, extrusion printing has greater versatility, both in the choice of printing materials and substrates. In view of the increasing structural complexity of current flexible electronics, extrusion printing also has apparent advantages in realizing high-precision conformal printing and multi-module integrated manufacturing, avoiding time-consuming and cumbersome transfer and assembly processes. More detailed comparisons of these two printing techniques can also be found in another of our reported works (*Nat. Commun.* 10, 1795 (2019))⁵, where we applied these two techniques to fabricate printed MXene micro-supercapacitors. The choice of these printing technologies is one of the keys to the success of printed electronics. In our revised manuscript, we accordingly added a comparison of these two printing technologies and a more detailed explanation of our choice of extrusion printing in this work, which can also be found below:

Supplementary Fig. 4 | Schematic illustration of the high-precision robotic deposition system for the direct printing of MXene inks. The three-axis mechanical system is capable of carrying the pneumatic extrusion nozzle according to the preset program to create conductive patterns on specific substrate surfaces accurately. The printing on curved surfaces is performed with the help of a computer-controlled image recognition system.

Unlike conventional subtractive manufacturing (such as photolithography), printing-based additive manufacturing is cost-effective for the rapid, large-scale production of flexible electronics due to its relatively simple procedures and desirable material utilization. Among various printing methods, extrusion-based direct ink printing offers greater opportunities for ink material selection and printing extensibility from micro to macroscale, plane to three-dimensional. Take the comparison with inkjet printing as an example, extrusion printing has apparent advantages in ink viscosity, printing throughput, and film thickness, as mentioned in our previous property-process guidelines summarized for MXene processing. Extrusion printing can directly deposit the continuous viscoelastic ink filaments without additional masks and accessories, which is a versatile method to realize functional patterns on different substrates (whether flat or curved) under ambient conditions. When considering the increasing structural complexity of flexible electronics, extrusion printing also has advantages in achieving high-precision conformal printing and multi-module integrated manufacturing to avoid

time-consuming cumbersome transfer and assembly processes. More detailed comparisons of these two printing techniques can also be found in another of our reported work. On the other hand, our extrusion printable MXene inks have high FoM values to achieve high-efficiency printing. Based on the above advantages, we chose the combination of extrusion printing and additive-free MXene inks to manufacture flexible wireless electronics and also further demonstrated the great potential of this approach in this field.

2. The use of additive-free MXene inks for printed electronics was first proposed in E. Quain, et al, Direct Writing of Additive-Free MXene-in-Water Ink for Electronics and Energy Storage, *Advanced Materials Technologies*, 4 (1) 1800256 (2019), not in Ref. 16.

Response: Thanks so much for this careful reminder. This (*Adv. Mater. Technol.* 4, 1800256 (2019))¹⁰ is a good work using direct writing (based on a rollerball pen) to fabricate conductive patterns and supercapacitors, initially showing the great potential of MXene ink for printed electronics. So, we have cited this article in our revised manuscript as an important reference for the development of additive-free MXene inks. In recent years, we have also reported multiple works on additive-free MXene inks for printed electronics. In 2018, our group reported the first work on MXene printed electronics (based on stamping) using additive-free MXene inks (*Adv. Funct. Mater.* 28, 1705506 (2018))⁴, this report quickly initiates widely research interests on printing of MXene inks in various applications, triggering many following up studies. In 2019, we further reported a more detailed work showing the application of additive-free MXene inks in printed electronics through inkjet printing and screen printing (*Nat. Commun.* 10, 1795 (2019))⁵. In 2020, we further demonstrated that, the efficient high resolution printing is not just confined to exfoliated MXene flakes inks, but also expanding to those MXene trash sediments, enriching of unetched MAX phase and unexfoliated Multilayered MXene particles—just by putting a quite small amount of Delaminated MXene sheets as conductive binders (*Adv. Mater.* 32 (17), 2000716 (2020))⁶. The printing and coating of MXene inks are further studied

(*ChemElectroChem* 8 (10), 1911-1917 (2021))¹¹, reviewed (*J. Phys.: Energy* 2 (3), 031004 (2020))¹², and perspected (*Mater. Today* 48, 214-240 (2020))³ by our group.

3. The authors claim stability of the printed patterns. Have they measured the conductivity of the printed patterns as a function of time? Water adsorption and swelling of the films may affect the electrical characteristics of MXenes.

Response: Many thanks for this important comment. The effect of water molecules on the electrical conductivity of printed MXene films is a non-negligible factor as it changes the interlayer spacing between nanosheets. But this effect is reversible, as the water molecules do not cause damage to the nanosheets themselves. Therefore, there have been many reports on using MXene as a humidity-sensitive material to develop humidity sensors. The main factor responsible for irreversible effects on MXene electrical conductivity is the oxidation of nanosheets. This process is particularly evident in aqueous solutions, as the dissolved oxygen in the water causes the MXene nanosheets to degrade from the edges, eventually forming cloudy-white colloidal solutions containing primarily anatase (TiO₂). To slow/avoid this oxidation process, the common treatments of MXene aqueous solution are argon protection and refrigerator storage. For printed MXene film, high-temperature annealing or low-humidity treatment is often used. On the other hand, recent research has pointed out that when MXene flakes are assembled and restacked, the freestanding MXene films retain their high electrical conductivity, and the oxidation is mitigated (*Adv. Funct. Mater.* 28, 1803360 (2018))¹³. Besides, the Gogotsi group reported the printed MXene patterns could keep chemical stability and electrical properties after storing for six months under ambient conditions (*Small* 17, 2006376 (2020))⁹. Accordingly, our experiments investigated the long-term stability of printed MXene films under ambient conditions and obtained similar results as previously reported. The results show that both unencapsulated and PDMS-encapsulated printed MXene films well retain their properties over one month. The specific experimental content and results are as follows:

Supplementary Fig. 14 | Long-term stability of printed MXene films under ambient conditions. **a**, Photograph of PDMS-encapsulated and unencapsulated printed MXene films. **b**, The resistance change of the printed MXene films in one month under ambient conditions. After printing, printed MXene films assembled and re-stacked from MXene flakes can maintain their high conductivity and mitigate oxidation. Both PDMS-encapsulated and unencapsulated printed MXene films maintain their properties well under conditioned storage for one month.

4. Schematic 1 in SI implies the orientation of MXene flakes parallel to the substrate surface. Is this really the case? Does it depend on the ink viscosity and printing parameters? The authors should provide information about orientation of flakes and SEM images of the printed line cross-section and surface, as flakes orientation will affect electronic transport and other properties.

Response: Many thanks for this professional comment. After printing, the microscopic arrangement and distribution of printed materials are important research objects. MXenes have typical morphological characteristics of 2D nanomaterials, usually reaching micron-scale flake diameters. As a result, under the effect of the shear force during printing, MXene flakes prefer to align in an orientation parallel to the substrate. In other words, MXenes have excellent film-forming properties (whether by printing or filtering), as demonstrated by much previous work (e.g., *Adv. Mater.*, 32, e2001093 (2020); *ACS Nano* **15**, 8860-8869 (2021))^{14,15}. In our previous study (*Adv. Mater.* 29, 1702678 (2017)), we used XRD and SEM to investigate in detail the

alignment relationship between the MXene flakes and the substrate after printing¹⁶. The conclusions we draw are consistent with the above statement: MXene flakes prefer to align parallel to the substrate after printing, and ordered flake arrangement can yield dense, highly conductive MXene films. Based on these references and experience, we studied this content again in the present work of using extrusion printing MXene inks to fabricate flexible electronics and came to similar conclusions. In our revised manuscript, we have made the following changes accordingly:

"Due to the shear force created in the nozzle of the printing head, $\text{Ti}_3\text{C}_2\text{T}_x$ nanosheets align themselves in the axial direction of the print head, forming highly-ordered liquid crystals within the extruded filaments and resulting in MXene films with densely stacked flakes in parallel to the substrate. Consequently, a high metallic electrical conductivity is enabled in the extrusion-printed MXene tracks. This result was also confirmed in subsequent experiments on the printed MXene films, such as XRD, SEM, and electrical measurements." (Supplementary Fig. 1)

"On the other hand, the high intensity (002) peak in the MXene XRD pattern also indicates that the $\text{Ti}_3\text{C}_2\text{T}_x$ flakes tend to align in an orientation parallel to the substrate under the effect of shear force and their own 2D material properties." (Supplementary Fig. 2a)

Supplementary Fig. 15 | Microscopic surface morphology of the printed MXene film. a, A cross-sectional SEM view of the printed MXene film. Scale bar, 20 μm . b, SEM image shows

densely stacked MXene flakes on the surface of the printed MXene film. Scale bar, 500 nm. The cross-section and top view demonstrate the continuous coverage of the printed MXene film on the substrate without protruding flake corners. Due to the shear force during extrusion printing and the 2D material properties of MXene, the $\text{Ti}_3\text{C}_2\text{T}_x$ flakes prefer alignment parallel to the substrate after printing, resulting in dense and highly conductive films. According to recent research, the fast charge transport within the printed MXene films is dominated by the intrinsic intra-flake processes.

5. The authors demonstrate printing on a variety of substrates. How does ink spreading (bleeding) depend on the substrate?

Response: We would like to thank the reviewer for raising this important question about the dispersion of MXene ink on the substrate. It is well-known that choosing a suitable substrate is one of the keys to successful printing. For high-resolution MXene ink extrusion printing, substrates with appropriate properties are quietly crucial, especially the wettability of the substrate has a significant influence on the morphology of printed patterns. In general, the substrate surface is flat and smooth, as demonstrated by the AFM measurements that commonly-used glass, polyethylene terephthalate (PET), and polyimide (PI) substrates generally have nano-level smooth surfaces (Supplementary Fig. 5). However, hydrophobic surfaces are challenging to print uniformly with aqueous inks due to the ink's tendency to retract and aggregate. Therefore, we introduced the plasma treatment as an efficient method of substrate pre-treatment to change the surface energy and improve the wetting properties of MXene aqueous ink on substrates. After plasma treatment, the decrease of the water contact angle of three commonly-used polymer substrates indicates the improved wetting properties (Supplementary Fig. 6), which contributes to the formation of continuous films and the adhesion enhancement between the ink and substrates. In Supplementary Fig. 7, we further present the contact angle between MXene ink and PET substrate with different plasma treatment time. Consequently, we selected 5 min of plasma time

for the substrate treatment, based on the optimization of surface energy matching and printing line width. In order to make the article as concise and understandable as possible, we mainly expand this part in detail in the Supplementary Material. A more detailed discussion can be found in Supplementary Figs. 5-7 and Supplementary Notes.

6. Very recently (already after submission of the current manuscript), a related paper was published in *Nature Electronics* by V. Nicolosi and F. Torrisi groups: E. Piatti et al., Charge transport mechanisms in inkjet-printed thin-film transistors based on two-dimensional materials, *Nature Electronics*, 4, 893–905 (2021). It discusses the mechanism of charge transfer in printed MXene films and may help with the explanation of the results reported in this work.

Response: Thanks so much for this professional comment. It is well known that metallic conductivity is a distinguishing feature of MXenes compared to other 2D nanomaterials. Therefore, this is an important and meaningful work (*Nat. Electron.* 4, 893-905 (2021))¹⁷ to elucidate the charge transport mechanisms in printed MXene films and provide theoretical guidance for the reliable design of more complex printed electronics using 2D inks. We read and studied this article carefully and agree with the conclusions drawn by the authors; that is, charge transport within the printed MXene films is dominated by intra-flake processes and mirrors that in the isolated constituent flakes. Accordingly, we have cited this article both in the manuscript and Supplementary Material. Below are the according changes we have made in our revised manuscript:

"Due to the shear-induced alignment during extrusion, the as-printed paths are composed of densely stacked and interconnected $\text{Ti}_3\text{C}_2\text{T}_x$ nanosheets (Fig. 2h and Supplementary Fig. 15), forming a robust metallic network that allows free and fast electron transport (dominated by the intrinsic intra-flake processes), thereby enabling high conductivity and mechanical flexibility (as evidenced by the cyclic bending test, Supplementary Figs. 16)." (Page 6 in manuscript)

"According to recent research, the fast charge transport within the printed MXene films is dominated by the intrinsic intra-flake processes." (Supplementary Fig. 15)

7. The authors should minimize the use of superlatives, such as "ideal", etc. There is no ideal ink – there is always room for further improvement.

Response: Many thanks for this kind reminder. The use of objective descriptors in manuscripts is an important principle worth following. Although we are excited by the excellent performance of the developed MXene inks in flexible electronics applications, we should also avoid using these overly descriptive terms. In our present manuscript, we have carefully checked and revised these words.

Finally, we would like to thank the reviewer again for these valuable comments and for the thoughtful and careful review towards improving our manuscript.

References

- 1 C. F. Zhang, S. J. Kim, M. Ghidui, M. Q. Zhao, M. W. Barsoum, V. Nicolosi and Y. Gogotsi, *Adv. Funct. Mater.* **26**, 4143-4151 (2016).
- 2 C. J. Zhang, S. Pinilla, N. McEvoy, C. P. Cullen, B. Anasori, E. Long, S.-H. Park, A. Seral-Ascaso, A. Shmeliov, D. Krishnan, C. Morant, X. Liu, G. S. Duesberg, Y. Gogotsi and V. Nicolosi, *Chem. Mater.* **29**, 4848-4856 (2017).
- 3 S. Abdolhosseinzadeh, X. Jiang, H. Zhang, J. Qiu and C. Zhang, *Mater. Today* **48**, 214-240 (2021).
- 4 C. Zhang, M. P. Kremer, A. Seral-Ascaso, S.-H. Park, N. McEvoy, B. Anasori, Y. Gogotsi and V. Nicolosi, *Adv. Funct. Mater.* **28**, 1705506 (2018).
- 5 C. J. Zhang, L. McKeon, M. P. Kremer, S. H. Park, O. Ronan, A. Seral-Ascaso, S. Barwich, C. O. Coileain, N. McEvoy, H. C. Nerl, B. Anasori, J. N. Coleman, Y. Gogotsi and V. Nicolosi, *Nat. Commun.* **10**, 1795 (2019).
- 6 S. Abdolhosseinzadeh, R. Schneider, A. Verma, J. Heier, F. Nuesch and C. J. Zhang, *Adv. Mater.* **32**, e2000716 (2020).
- 7 G. C. Pidcock and M. in het Panhuis, *Adv. Funct. Mater.* **22**, 4790-4800 (2012).
- 8 X. Tang, G. Murali, H. Lee, S. Park, S. Lee, S. M. Oh, J. Lee, T. Y. Ko, C. M. Koo, Y. J. Jeong, T. K. An, I. In and S. H. Kim, *Adv. Funct. Mater.* **31**, 2010897 (2021).
- 9 S. Uzun, M. Schelling, K. Hantanasirisakul, T. S. Mathis, R. Askeland, G. Dion and Y. Gogotsi, *Small* **17**, 2006376 (2020).
- 10 E. Quain, T. S. Mathis, N. Kurra, K. Maleski, K. L. Van Aken, M. Alhabeab, H. N. Alshareef and Y. Gogotsi, *Adv. Mater. Technol.* **4**, 1800256 (2019).
- 11 S. Abdolhosseinzadeh, J. Heier and C. F. Zhang, *ChemElectroChem* **8**, 1911-1917 (2021).
- 12 S. Abdolhosseinzadeh, J. Heier and C. F. Zhang, *J. Phys.: Energy* **2**, 031004 (2020).
- 13 G. M. Weng, J. Y. Li, M. Alhabeab, C. Karpovich, H. Wang, J. Lipton, K. Maleski, J. Kong, E. Shaulsky, M. Elimelech, Y. Gogotsi and A. D. Taylor, *Adv. Funct. Mater.* **28**, 1803360

- (2018).
- 14 J. Zhang, N. Kong, S. Uzun, A. Levitt, S. Seyedin, P. A. Lynch, S. Qin, M. Han, W. Yang, J. Liu, X. Wang, Y. Gogotsi and J. M. Razal, *Adv. Mater.* **32**, e2001093 (2020).
 - 15 J. H. Kim, G. S. Park, Y. J. Kim, E. Choi, J. Kang, O. Kwon, S. J. Kim, J. H. Cho and D. W. Kim, *ACS Nano* **15**, 8860-8869 (2021).
 - 16 C. F. Zhang, B. Anasori, A. Seral-Ascaso, S. H. Park, N. McEvoy, A. Shmeliov, G. S. Duesberg, J. N. Coleman, Y. Gogotsi and V. Nicolosi, *Adv. Mater.* **29**, 1702678 (2017).
 - 17 E. Piatti, A. Arbab, F. Galanti, T. Carey, L. Anzi, D. Spurling, A. Roy, A. Zhussupbekova, K. A. Patel, J. M. Kim, D. Daghero, R. Sordan, V. Nicolosi, R. S. Gonnelli and F. Torrioni, *Nat. Electron.* **4**, 893-905 (2021).

REVIEWERS' COMMENTS

Reviewer #1 (Remarks to the Author):

The authors have reflected the comments of the reviewers, and the manuscript has been improved over the previous version. Therefore, it is recommended to publish without further revision.

Reviewer #2 (Remarks to the Author):

I'm satisfied with the revised version and responses of the authors to both reviewers.

Responses to Reviewers' Comments:

First of all, we would like to thank the reviewers for their time and effort in the first round of review. The reviewers' thoughtful comments and suggestions have resulted in a noticeable improvement in the quality and clarity of our manuscript. We are grateful to the reviewers for their comments on our manuscript. Our point-to-point responses in the second-round of review are presented below.

Reviewer #1 (Remarks to the Author):

The authors have reflected the comments of the reviewers, and the manuscript has been improved over the previous version. Therefore, it is recommended to publish without further revision.

Response: We are delighted to see your positive comments. Thank you so much for reviewing our article. Your insightful comments greatly improve and strengthen our work.

Reviewer #2 (Remarks to the Author):

I'm satisfied with the revised version and responses of the authors to both reviewers.

Response: Thank you so much for your kindness and assistance in improving our work. We are honored to have your approval of our revised article.